# Enhancing drought monitoring and assessment capability in India through high-resolution (250m) data

3 Anukesh Krishnankutty Ambika<sup>1</sup> and Vimal Mishra<sup>1,2</sup>

4

7

- 5 <sup>1</sup>Earth Sciences, Indian Institute of Technology (IIT) Gandhinagar
- 6 <sup>2</sup>Civil Engineering, Indian Institute of Technology (IIT) Gandhinagar
- 8 Correspondence to: vmishra@iitgn.ac.in

# 9 Abstract

- Drought poses a tremendous challenge to India's socioeconomic development, livelihood,
- agriculture, and water management. While existing drought monitoring systems have
- characterized drought impact at different scales, policymaking and management require
- drought assessment at sub-district or taluka (sub-district) levels. Here, we develop high-
- resolution (250 m) agriculture drought indices for the Indian region to overcome the
- shortcomings of the coarse resolution datasets. We used the co-kriging to downscale the Land
- Surface Temperature (LST) from 1000m to 250m. The LST and Enhanced Vegetation Index
- (EVI) are obtained at 8-day intervals at 250m spatial resolution. The high-resolution datasets
- show significant improvement in identifying the severity and coverage of drought. Soil
- Moisture Agriculture Drought Index (SMADI), which accounts for water stress and
- vegetation lag response, shows high reliability in drought detection. We evaluated drought
- extent and severity using the newly developed dataset and found that the high-resolution
- dataset can be used to separate the irrigation impact on drought alleviation. The high-
- resolution drought indices from SMADI and the Normalized Vegetation Supply Water Index
- (NVSWI) effectively represent the drought conditions at district and taluka levels that can be
- used in drought impacts assessments in India.

# 27 1 Introduction

- Drought is one of the complex natural hazards (Lloyd-Hughes, 2014; Van Loon, 2015;
- Wilhite et al., 2000), which poses tremendous challenges to water resources management,
- agriculture, and Gross Domestic Product (GDP) due to a sustained deficit of water
- availability (Godfray et al., 2010; Mooley and Parthasarathy, 1983; Wilhite, 2005). The
- vulnerability of the Indian population to drought is relatively high due to economic viability
- from the agriculture sector (Mishra and Singh, 2010). The recent increase in temperature and

| 35 | erratic summer monsoon have impacted the frequency, intensity, and areal extent of drought       |  |  |  |  |
|----|--------------------------------------------------------------------------------------------------|--|--|--|--|
| 36 | over the Indian region (Mishra et al., 2012; Roxy et al., 2015). Moreover, the frequency of      |  |  |  |  |
| 37 | flash drought has also increased in recent decades (Mahto and Mishra, 2020). For example,        |  |  |  |  |
| 38 | ten major droughts occurred between 1950 and 1989, while five occurred after 2000 (Mishra,       |  |  |  |  |
| 39 | 2020). The frequency of flash drought is projected to increase seven-fold by the end of the      |  |  |  |  |
| 40 | 21st century, with a considerable economic implications (Mishra et al., 2021). For instance,     |  |  |  |  |
| 41 | the 2014-2015 drought resulted in the loss of billions of dollars affecting more than 3.3        |  |  |  |  |
| 42 | million people in India (Mishra et al., 2018). Therefore, quantifying drought impacts at high    |  |  |  |  |
| 43 | resolution is necessary for water management and food security.                                  |  |  |  |  |
| 44 |                                                                                                  |  |  |  |  |
| 45 | Meteorological, agricultural, hydrological, and socioeconomic droughts (Wilhite and Glantz,      |  |  |  |  |
| 46 | 1985) are propagated and intensified through land-atmospheric interactions, local land           |  |  |  |  |
| 47 | surface characteristics, soil moisture availability, regional climate change, and human          |  |  |  |  |
| 48 | interferences (Barker et al., 2016; Van Loon and Laaha, 2015; Mishra et al., 2021; Shah et       |  |  |  |  |
| 49 | al., 2021). The temporal characteristics, area affected, extent, frequency, severity, intensity, |  |  |  |  |
| 50 | and duration of drought are characterized by several drought indices (Dai, 2011; Mishra et       |  |  |  |  |
| 51 | al., 2016; Yu et al., 2014). Drought characteristics are monitored using Standardized            |  |  |  |  |
| 52 | Precipitation index [SPI; (McKee et al., 1993)] and Standardized Precipitation                   |  |  |  |  |
| 53 | Evapotranspiration Index [SPEI; (Vicente-Serrano et al., 2010)], which incorporate the           |  |  |  |  |
| 54 | influence of precipitation, temperature, and evapotranspiration on drought estimates.            |  |  |  |  |
| 55 | Moreover, the Palmer drought Severity Index (PDSI) takes into account soil water balance to      |  |  |  |  |
| 56 | identify drought by considering the potential loss of moisture due to temperature (Palmer,       |  |  |  |  |
| 57 | 1965). Further, Standardized Soil Moisture Index [SSMI; (Hao and AghaKouchak, 2013)]             |  |  |  |  |
| 58 | and Standardized Streamflow Index [SSI; (Bhardwaj et al., 2020)] are widely used for             |  |  |  |  |
| 59 | agricultural and hydrological droughts, respectively. Although these indices may provide         |  |  |  |  |
| 60 | valuable information on drought, high uncertainties exist in drought detection due to sparse     |  |  |  |  |
| 61 | weather stations and spatial interpolation.                                                      |  |  |  |  |
| 62 |                                                                                                  |  |  |  |  |
| 63 | Several drought monitoring, warning, and prediction related measures are relatively less         |  |  |  |  |
| 64 | developed than the other major disasters due to the complexity of the process involved in        |  |  |  |  |
| 65 | identifying and propagating drought (Saha et al., 2021). Drought impact assessment-related       |  |  |  |  |
| 66 | efforts in India are limited due to a lack of fine-scale/higher resolution information that can  |  |  |  |  |

resolve sub-district level characteristics (Shah and Mishra, 2015). For instance, the near real-

| time drought monitoring in South Asia at 0.05° indicated that the bias-corrected high-                |  |  |  |  |  |
|-------------------------------------------------------------------------------------------------------|--|--|--|--|--|
| resolution datasets effectively capture observed drought variability, similar to information          |  |  |  |  |  |
| obtained by satellite remote sensing (Aadhar and Mishra, 2017). The near real-time drought            |  |  |  |  |  |
| system for the Indian region considers meteorological information (Shah and Mishra, 2015).            |  |  |  |  |  |
| India Meteorological Department (IMD) provides monthly scale drought information at                   |  |  |  |  |  |
| relatively coarser resolution ( <u>www.imdpune.gov.in</u> ), which is helpful for the decision making |  |  |  |  |  |
| at the administrative level (district). Furthermore, satellite-based near real-time drought           |  |  |  |  |  |
| monitoring and early warning systems provide a drought warning at the state level (Takeuchi           |  |  |  |  |  |
| et al., 2015). Bias corrected high-resolution near real-time drought monitoring at 0.05°              |  |  |  |  |  |
| provides the severity of drought over South Asia (Aadhar and Mishra, 2017). While the                 |  |  |  |  |  |
| existing drought monitoring system in the Indian region offers important information on               |  |  |  |  |  |
| drought, decision-making at the local level is hindered due to their coarse spatial resolution.       |  |  |  |  |  |
| Therefore, remote sensing-based high-resolution drought monitoring can be used as a                   |  |  |  |  |  |
| supplement to garner the spatial variability of drought impact.                                       |  |  |  |  |  |
|                                                                                                       |  |  |  |  |  |
| Vegetation indices are commonly used satellite-based drought monitoring at high resolution            |  |  |  |  |  |
| (Bannari et al., 1995). Moreover, vegetation stress indices incorporating ecosystem                   |  |  |  |  |  |
| components are more prominent for drought detection (Jiao et al., 2021). Although the                 |  |  |  |  |  |
| vegetation stress alone can indicate drought onset and termination (Agutu et al., 2017),              |  |  |  |  |  |
| combining land surface temperature improves the drought prediction due to the changes in              |  |  |  |  |  |
| local biophysical (soil, slope) and climate conditions (García-León et al., 2019). Moreover,          |  |  |  |  |  |
| the additive impact of surface temperature and vegetation stress is highly correlated with the        |  |  |  |  |  |
| crop yield in various agro-meteorological zones (Kogan et al., 2012; Prasad et al., 2006;             |  |  |  |  |  |
| Rahman et al., 2009). Since agricultural drought is modulated by the land surface condition,          |  |  |  |  |  |
| separating irrigation impact on the cropping area is crucial for identifying the drought extent       |  |  |  |  |  |
| (Mishra et al., 2016) as irrigation modulates the vegetation health and surface temperature           |  |  |  |  |  |
| during the summer (Ambika and Mishra, 2019). In addition, various vegetation-related                  |  |  |  |  |  |
| remote sensing drought indices that combine surface temperature with vegetation conditions            |  |  |  |  |  |
| can be a viable indicator in monitoring agricultural drought (Bento et al., 2018; Gomes et al.,       |  |  |  |  |  |
| 2017; Rojas et al., 2011). High-resolution drought monitoring at a regional scale can also be         |  |  |  |  |  |
| valuable for decision making at sub-district (Taluka) levels.                                         |  |  |  |  |  |
|                                                                                                       |  |  |  |  |  |
|                                                                                                       |  |  |  |  |  |

- $\label{eq:lambda} \mbox{ Land surface temperature (LST) is one of the critical parameters for an integrated high-$
- resolution drought monitoring system since it indirectly measures surface energy balance

- (Tomlinson et al., 2011). Thermal stress is a good indicator for early drought detection,
- derived from LST (Anderson et al., 2008; Seyednasrollah et al., 2019). The combination of
- LST and EVI indices can be an excellent indicator for multi-sensor drought detection and
- monitoring strategies (Orhan et al., 2014). The relation between thermal stress and vegetation
- condition has been successfully applied for drought monitoring (Seyednasrollah et al., 2019).
- Further, while combining with other metrics like soil moisture, the LST-EVI relationship has
- shown potential for improved drought monitoring (Hao et al., 2015; Jiao et al., 2019).
- We develop a high-resolution drought index using LST and EVI at 250 m resolution. We
- developed Vegetation Health Index (VHI), Vegetation Condition Index (VCI), Temperature
- Condition Index (TCI), Normalized vegetation Supply Water Index (NVSWI), and Soil
- Moisture Agriculture Drought Index (SMADI) at 250 m. Moderate Resolution Imaging
- Spectroradiometer (MODIS) datasets were used to develop eight-day continuous LST and
- enhanced vegetation index (EVI). The high-resolution agriculture drought dataset at 250 m
- resolution at the national scale can be used for impact assessment.

#### 114 2 Methods

# 115 2.1 Enhanced Vegetation Index (EVI) at 8-day interval

The Enhanced Vegetation Index (EVI) can identify the variation in leaf area index (LAI), 116 canopy cover, and photosynthetically active radiation (Gao et al., 2000). Therefore, EVI is 117 118 useful in monitoring seasonal, inter-annual, and inter-annual long-term variation in vegetation stress (Huete et al., 2002). Moreover, the blue wavelength corrections for distortion make 119 EVI not saturate quickly, as is the case of the Normalized Difference Vegetation Index 120 (NDVI) [Gao et al., 2000]. Further, EVI is sensitive to the green biomass response in varying 121 122 weather conditions. EVI from MODIS provides global coverage at a sixteen-day interval with a better spectral, spatial, geometric, and radiometric quality (Didan et al., 2015). Moreover, 8-123 124 day EVI can detect vegetation response to changes in atmospheric vapour pressure deficit, clouds, and sun view angles (Gurung et al., 2009). We developed the 8-day MODIS EVI 125 126 temporal composite at 250 m for the 2000-2017 period. We used daily MOD09Q1 [Red (620-720 nm) and Near Infrared (841-876 nm)] at 250 m and MOD09A1 (Blue 459-479 nm) at 127 128 500 m surface reflectance. The MOD09A1 Band-3 is resampled using the nearest neighbour to keep the spatial consistency of the raw dataset. The eight-day composite is derived from 129 the datasets corrected for atmospheric conditions like aerosol, Rayleigh scattering, and 130 131 gasses. EVI at 250 resolution is obtained using the same algorithm provided for the EVI

- (Didan et al., 2015). The abbreviation and the summary of datasets used in the study are
- given in Tables 1 and 2.

### 135 2.2 Downscaling Land Surface Temperature (LST) data at 250 m

- There have been numerous satellite LST observations in recent decades with limited spatial
- and temporal resolution (Gutman, 1999; Li et al., 2014), restricting their use to broader
- hydrological applications. For example, the National Oceanic and Atmospheric
- Administration (NOAA) Star Center for Satellite Application and Research (NSTAR)
- provide weekly LST at 4 km spatial resolution from 1982 to 2018 (Tomlinson et al., 2011).
- However, ASTER satellite data at 90 m spatial resolution revisit the same area every 16 days.
- Therefore, a high-resolution (spatial and temporal) LST dataset adds value to drought
- monitoring.
- We downscaled MODIS (MOD11A2) LST 1000 m to 250 m using the co-kriging method
- (Pardo-Igúzquiza et al., 2006). The downscaled LST was then combined with EVI to evaluate
- various drought indices over India. The 8-day MODIS data product MOD11A2 land surface
- temperature (LST) corresponds to an average value for the period. The improvement in
- version 6 of the MODIS LST uses a split-window algorithm with comprehensive regression
- analysis, reducing LST uncertainties' sensitivity (Wan, 2006). All the MODIS granules over
- the Indian region were mosaicked and reprojected to the geographic coordinates system using
- the NASA reprojection tool (mrtweb.cr.usgs.gov).

Downscaling combines two or more data sets of different spatial resolutions to derive an

- enhanced resolution dataset (Pardo-Iguzquiza et al., 2011). Previous studies have used
- empirical relations between visible, near-infrared, and shortwave infrared (SWIR) bands and
- Vegetation Index (NDVI or EVI) for high resolution (Agam et al., 2007; Gowda et al., 2007;
- Jeganathan et al., 2011; Nichol and Wong, 2005). However, downscaling provides promising
- results since it preserves the variation of ground features and maintains image geometry
- coherence (Rodriguez-Galiano et al., 2012). The correlation between LST and spectral bands
- is low (Rodriguez-Galiano et al., 2012). However, a joint variability pattern can be observed
- between the LST and the spectral bands (Drury, 1987). Further, LST can be downscaled
- using the joint variability of the cross-covariance (Liu et al., 2006). We used co-kriging as an
- approximation method with the high-resolution data to downscale the LST (Stathopoulou and
- Cartalis, 2009). Previous studies using experimental cross-covariances and direct covariances

considers the pixel size and the sensor's point-spread function to calculate the weights for 165 downscaling, which is an added advantage compared to other methods (Kustas et al., 2003). 166 167 Hence, the downscaled image preserves the spatial and radiometric variability (Rodriguez-168 Galiano et al., 2012). Further, the cokriging ensures identical spatial variability of the raw datasets even when the point-scale function degrades the spatial coherence (Liu et al., 2006; 169 170 Rodriguez-Galiano et al., 2012; Stathopoulou and Cartalis, 2009). Downscaling of LST is processed with EVI and Shuttle Radar Topography Mission (SRTM) 171 elevation datasets as covariates. The SRTM elevation is resampled with cubic convolution at 172 173 250 m to maintain spatial consistency. Since the elevation is one of the prominent factors in changing the land surface temperature, we used SRTM elevation as another covariate. 174 175 Further, the Indian subcontinent is divided into 1200 tiles, with each tile covering an area coverage of approximately 0.34 million hectares (mha). The majority of tiles are confined to 176 an individual agro-ecological zone. The downscaling weights were calculated from both 177 covariates to downscale LST at 250 m. 178

showed promising results in downscaling Landsat LST (Agam et al., 2007). The Cokriging

The downscaled LST was evaluated against 1km LST using structural similarity index

[SSIM; (Wang et al., 2004)]. SSIM evaluates image quality based on luminescence, contrast,

and structural differences between the degraded (high resolution) image and the original

image (low resolution). SSIM ranges between -1 and 1, with values closer to 1 showing better

similarity (Rodriguez-Galiano et al., 2012). The image quality index (IQI) was also used

(Wang and Bovik, 2002) to account for luminance distortion, loss of correlation, and contrast

distortion [Table S2]. The quality of the downscaled data was evaluated for different regionsin India using districts and talukas boundaries.

#### 188 2.3 High-Resolution vegetation Indices

We calculated various agriculture drought indices from the downscaled LST and EVI at 250

190 m. First, we obtained the Vegetation Condition Index (VCI), which indicates the vegetation

stress and is the most commonly used agriculture drought index (Kogan, 1995a). VCI can

- isolate the weather-related vegetation stress and detect the drought onset, intensity, and
- impact on vegetation (Kogan, 1995a). Unlike VCI, Temperature Condition Index (TCI)
- determines the vegetation stress caused by temperature and excessive wetness. We calculated

effective indicators for drought detection, combining both indices could be more effective in
determining the drought intensity (Kogan, 1995a; Rojas et al., 2011). For example, the
Vegetation Health Index (VHI) is an additive combination of VCI and TCI for drought
detection. Moreover, Kogan (1995) proposed VHI to remove cloud effects from the
Advanced Very High-Resolution Radiometer (AVHRR) thermal band (Kogan, 1995a,

both indices for the 2000-2017 period (Kogan, 1995a). Even though the VCI and TCI are

- 1995b). Therefore, VHI indicates drought for seasons having high temperatures and
- favourable conditions for low temperatures.

Soil moisture plays a crucial role in drought detection and identification (Seneviratne et al., 203 204 2010). Integrating soil moisture in drought indices enhances our understanding of land-205 atmospheric interaction in modulating the drought event (Seneviratne et al., 2010). We 206 calculated the Soil Moisture Agricultural Drought Index (SMADI) by combining surface temperature conditions, lagged response of vegetation, and soil moisture to detect the short-207 term drought (Sánchez et al., 2016). The SMADI can provide early warming of yield 208 209 reduction due to its sensitivity to water stress (Souza et al., 2021). Surface soil moisture for the SMADI index is obtained from the Global Land Evaporation Amsterdam Model 210 (GLEAM; 0.25°) at a 10 cm depth and resampled at a resolution of 250 m. We used the 211 nearest neighbour resampling method to keep the spatial consistency with the original 212 213 dataset. The GLEAM v3.2a soil moisture uses extensive validation against the in-situ data points having higher accuracy than other data GLEAM v3.2b (Martens et al., 2017). To 214 215 compensate for the SMADI response towards drought, we calculated the Normalized vegetation Supply Water Index [NVWSI; (Abbas et al., 2014)]. The NVSWI assumes that 216 land surface temperature will be low when sufficient soil water supply exists (Abbas et al., 217 2014). However, during the dry condition, the leaf stomata are partly closed to sustain water 218 219 stress, resulting in a reduction in evapotranspiration and increased surface temperature (Zhou et al., 2019). Hence, the NVSWI depends on vegetation health and indirectly indicates the 220 221 soil moisture-induced drought changes. 222 We used the Standardized Evaporation Deficit Index [SEDI - 0.25°; (Vicente-Serrano et al., 223

- 2018) ]and Drought Severity Index  $[DSI 0.05^{\circ} \& 0.25^{\circ}; (Mu et al., 2013)]$  to evaluate the
- drought estimates from NVSWI and VHI for the Indian Region. The derived drought indices
- were aggregated to  $0.05^{\circ}$  using the majority resampling techniques to compare drought
- extent. We used ranges of indicators to categorize drought as incipient drought (between -0.5

| 228               | and -0.59), mild drought (between -0.6 and -0.89), moderate drought (between -0.9 and -           |  |  |  |  |
|-------------------|---------------------------------------------------------------------------------------------------|--|--|--|--|
| 229               | 1.19), severe drought (between -1.2 and -1.49), and extreme drought (between -1.5 and less).      |  |  |  |  |
| 230               |                                                                                                   |  |  |  |  |
| 231               | 3 Result and Discussion                                                                           |  |  |  |  |
| 232<br>233<br>234 | 3.1 Land Surface temperature at 250 m resolution                                                  |  |  |  |  |
| 235               | First, we evaluated the quality of the downscaled LST at 250m during February 2000 (Fig. 1).      |  |  |  |  |
| 236               | We observed that the high-resolution and coarse-resolution LST display similar SSIM and           |  |  |  |  |
| 237               | IQM over the selected region of central India (Fig. 1b,c). However, as expected, LST at 250       |  |  |  |  |
| 238               | m displays greater spatial details, useful for drought assessment (Fig. S2 & S3). The             |  |  |  |  |
| 239               | downscaled LST indicates geographic variability, considering using the SRTM elevation data        |  |  |  |  |
| 240               | (Fig. S3). To evaluate the spatial variability of drought, areas from diverse climatic settings   |  |  |  |  |
| 241               | were selected. Initially, the LST was downscaled using EVI as a covariate, indicating lesser      |  |  |  |  |
| 242               | SSIM (Fig. S2). Furthermore, downscaling LST by EVI and elevation dataset as a covariate          |  |  |  |  |
| 243               | improved the spatial dispersion coherently (Fig S2). The structural variability of LST            |  |  |  |  |
| 244               | enhanced significantly from single to multi covariate downscaling (Fig. S2 ). Moreover, by        |  |  |  |  |
| 245               | including multiple covariates, the co-kriging improved the coherence of the spatial continuity    |  |  |  |  |
| 246               | in downscaled LST (Rodriguez-Galiano et al., 2012). Further, we considered an area                |  |  |  |  |
| 247               | characterized by various natural land covers, with vegetation mixtures, build-up, cropping        |  |  |  |  |
| 248               | area, bare soils, and urban land area to evaluate the spatial variance in LST downscaling. All    |  |  |  |  |
| 249               | the downscaled images were identical to the original 1000m, indicating less bias in tone,         |  |  |  |  |
| 250               | contrast, and saturation [Fig. S4 & S5; Table. S1 & S2]. However, the downscaled image            |  |  |  |  |
| 251               | showed a consistent mean value of LST with variation in standard deviation. Our results           |  |  |  |  |
| 252               | show a good agreement between 250 m and 1000 m LST with a mean SSIM value of 0.52 for             |  |  |  |  |
| 253               | district and taluka boundary areas (Table S1). Further, the IQI shows less luminance and          |  |  |  |  |
| 254               | contrast distortion with a high correlation. Both district and taluka levels have a higher degree |  |  |  |  |
| 255               | of confidence between 250 m and 1000 m LST with a mean value of 0.86 (Table S2). The              |  |  |  |  |
| 256               | downscaled LST signifies variation in continuity as it is expected that high-resolution           |  |  |  |  |
| 257               | datasets represent higher spatial variability than low-resolution with lesser pixel numbers       |  |  |  |  |
| 258               | (Pardo-Iguzquiza et al., 2011; Pardo-Igúzquiza et al., 2006). In general, the downscaled LST      |  |  |  |  |
| 259               | can be used with EVI of the same resolution for monitoring the agriculture drought at 250         |  |  |  |  |
| 260               | m.                                                                                                |  |  |  |  |
| 261               |                                                                                                   |  |  |  |  |

# 262 **3.2 Drought assessment at different resolutions**

| 263 | Next, we estimated the area under drought for DSI and SEDI from 2000 to 2011. We                |
|-----|-------------------------------------------------------------------------------------------------|
| 264 | observed that drought impacts around 10% of the Indian region each year. Moderate-              |
| 265 | resolution (0.05°) DSI and low-resolution SEDI (0.25°) were analyzed to understand the          |
| 266 | variability in drought severity (Fig. 2a,b). As expected, DSI at 0.05° shows a reasonable       |
| 267 | improvement in capturing the spatial and temporal variability of drought-affected areas         |
| 268 | during 2000-2011. DSI integrates remotely sensed NDVI, potential evapotranspiration, and        |
| 269 | evapotranspiration (Mu et al., 2013). Moreover, DSI incorporates vegetation response to the     |
| 270 | dry condition and terrestrial water availability associated with dryness or wetness (Mu et al., |
| 271 | 2013). Further, we selected different drought-affected areas to evaluate the spatial extent of  |
| 272 | drought severity change at different resolutions. The eight-day dataset of DSI identified the   |
| 273 | mesoscale geographical variability of the severe drought period compared to the SEDI, which     |
| 274 | is available at a monthly scale (Fig. 2 c-f & g-j). We note DSI and SEDI follow a similar       |
| 275 | pattern of the area under drought (Fig. 2a, b). Since SEDI exhibits a higher correlation with   |
| 276 | the vegetation anomalies, SEDI identifies water stress sensitivity to leaf activity (Vicente-   |
| 277 | Serrano et al., 2018). Moreover, SEDI is formulated based on the evaporative deficit, which     |
| 278 | signifies a similar spatial extent of drought as DSI. The spatial severity of drought in DSI    |
| 279 | indicates that high-resolution datasets can improve the understanding of drought impacts. For   |
| 280 | example, the NDVI and LST at 250m can separate the drought impact in irrigated and rainfed      |
| 281 | areas (Ambika and Mishra, 2019). The drought severity analysis by combining model output        |
| 282 | with observation highlights the uncertainty in percentage area under drought (Aadhar and        |
| 283 | Mishra, 2017). Hence, the noticeable difference in extreme drought-impacted areas in DSI        |
| 284 | and SEDI emphasizes accounting for the spatial variability of drought.                          |
| 285 | We selected four areas highlighted during the significant drought period to quantify the        |

spatial variability of drought extent at high-resolution (Fig. 2 c-f). We compared 0.25° and 286 0.05° DSI with 250 m NVSWI and VHI (Fig. 3). The difference in the spatial variability of 287 drought shows the bias in drought extent at a coarser resolution (Fig. 3q). For instance, the 288 NVSWI and VHI show relatively low values for the drought extent during 2002 and 2005 289 compared to 2000 and 2009. Further, during 2009 the DSI underestimated drought extent by 290 45%. Based on the 12-month SPI and SPEI at 0.05°, drought analysis identifies 40-50% of 291 292 Bulandshahr district under severe drought during 2015 (Aadhar and Mishra, 2017). On the other hand, the same analysis at 0.05° eliminates the drought condition in other districts 293 [Faisalabad and Ratnapura; (Aadhar and Mishra, 2017)], which further indicates the utility of 294 high-resolution drought monitoring to identify the macroscale variability. 295

### 297 **3.3** Temporal variability in agriculture drought

Most of the Indian region underwent different drought events during the past decades. One of
the deadliest meteorological droughts lasted from 2000 to 2003 (Mishra, 2020). During 2002
the drought was caused by a precipitation deficit of 21.5% during the summer monsoon
season. Further, in July, a precipitation deficit of 56% had a devastating impact on the
socioeconomic environment (Mishra, 2020). Considering these, we used the 2002 summer as
a case study to evaluate the spatial pattern detected in the newly developed high-resolution
drought indices.

We identified drought during 2002 March and compared the spatial extent of all the indices. 306 DSI and SEDI show a similar drought extent and are more prominent over the Indo-Gangetic plain and Deccan plateau (Fig. S6). Further, irrigation-induced alleviation of agriculture 307 stress is not observed in the DSI and SEDI. For instance, Ambika et al. (2019) identified that 308 309 the vegetation stress on peak growing period is significantly reduced by irrigation. NVSWI 310 and VHI show similar vegetation stress changes along the Indo-Gangetic plain (Fig S6). the 311 Multi-index drought at 250m shows consistency in drought severity extent (Fig. 4). However, 312 the soil moisture-induced SMADI shows a more prominent impact of drought (Fig. 4e-h). 313 The existence of soil moisture and lagged response of agriculture stress in SMADI can characterize the drought condition, particularly in areas where crop yield is more sensitive to 314 water stress (Souza et al., 2021). We identified a similar extent of drought severity in NVSWI 315 and SMADI along the Indian region, indicating that soil moisture has lagged response to 316 317 vegetation stress [Fig. 5; (Gurung et al., 2009)]. Moreover, all the drought indices at 250 m show a similar pattern of spatial extent during September 2002. The comparisons of multi-318 index drought at 250m show that the areal extent and severity of drought at high-resolution 319 320 are necessary for management at the taluka (sub-district) level. 321 Finally, we developed district and taluka level maps of drought severity and the extent to 322 understand the applicability in assisting decision-making (Fig. 6). We considered the 2015 drought to evaluate the vegetation drought response. The meteorological drought in 2015 was 323 the longest in the entire record of a century and peaked in June 2016 (Mishra, 2020). The 324 325 2016 drought affected more than 16% of the country but was less severe than the other 326 meteorological drought (Mishra, 2020). The 8-day high-resolution drought obtained from the

327 NVSWI and SMADI shows that a large area of central India experienced severe and extreme

- drought (Fig. 6a, b). Further, the highlighted maps of district and taluka level under drought
  severity show the consistent extent in SMADI and NVSWI, which can be used for decision
- and policymaking (Fig. 6c-f).

# 332 4 Data availability

- The high-resolution LST and NVSWI are publicly available from the Zenodo versions link:
- <u>https://doi.org/10.5281/zenodo.6798442</u>. The dataset covers the Indian region at 8-day
- temporal resolution at 250 m spatial resolution for the 2000 2017 period. The dataset is
- provided in WGS 1984 projection and tiff format.

# 338 5 Conclusions

The current study presents a newly developed high-resolution land surface temperature and 339 340 enhanced vegetation index dataset at an 8-day interval with 250 m resolution over the Indian 341 region. Further, we developed different agriculture drought indices (VCI, TCI, VHI, NVSWI, and SMADI) at 250 m. The data is derived from satellite-based MODIS and GLEAM surface 342 343 soil moisture covering the entire Indian region from 2000 to 2017. The eight-day dataset is provided to facilitate characterization of drought severity and extent at high resolution. 344 Moreover, the increased frequency of drought monitoring helps to characterize agricultural 345 drought at high temporal resolution for the Indian region. The high-resolution drought indices 346 show significant improvement in detecting drought extent and severity. The multi-index 347 348 drought can characterize the drought impact at district and taluka (sub-district) boundaries. 349 The inclusion of soil moisture in SMADI accounts for the water stress, and lag response highlights drought severity. SMADI and NVSWI show high reliability in investigating 350 drought detection capability at the district and taluka levels. The high-resolution multi-index 351 drought can act as an early warning to drought detection and mitigation compared to the other 352 353 hydrological, meteorological and socioeconomic drought indices. The high-resolution dataset exhibits the potential to separate the land management impact on the drought alleviation-for 354 instance, the extensive irrigation in the Indo-Gangetic plain. These results highlight the 355 validity and advantage of high-resolution drought monitoring, and its unprecedently high 356 resolution offers critical benefits to monitoring and assessment for policy and decision-357 358 makers.

- Acknowledgement: Authors appreciate the Ministry of Human Resources Development 360
- (MHRD). The authors acknowledge the data availability from the Moderate Resolution 361
- Imaging Spectroradiometer (MODIS). Land Surface Temperature (LST), Enhanced 362
- Vegetation Index (EVI) and all the ancillary can be obtained from 363
- https://e4ftl01.cr.usgs.gov/. The Global Land Evaporation Amsterdam Model (GLEAM) soil 364
- moisture can be obtained from https://www.gleam.eu/#downloads . The global gridded
- monthly Standardized Evaporation Deficit Index (SEDI) and Drought Severity Index (DSI)
- can be downloaded from https://digital.csic.es/handle/10261/160091 and 367
- http://files.ntsg.umt.edu/data/NTSG Products/DSI/ (https://larsjung.de/h5ai/) (umt.edu). The 368
- global map of irrigated area version 5 was collected from 369
- https://www.fao.org/aquastat/en/geospatial-information/global-maps-irrigated-areas/latest-
- version/. The global irrigated temporal map of fractional Historic Irrigation Datasets can be obtained from https://mygeohub.org/publications/8/2.

Code Availability: Codes used to develop the high-resolution datasets can be obtained from 374 the corresponding author.

- Competing interests: Authors declare no competing interest.
- Author contributions: VM and AKM designed the study. AKA developed the dataset and 376 wrote the initial draft. VM and AKA discussed results and enhanced the initial draft. 377

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
