# Peer review of "Enhancing drought monitoring and assessment capability in India through high-resolution (250m) data"

_Earth System Science Data, 2022_

## Referee Comment (RC2)

Review for "Global seasonal-scale meteorological drought events and concurrent temperature abnormalities: Detection, classification, and characteristics of process evolution" in *Earth System Science Data (ESDD)*

RECOMMENDATION: *Rejection and resubmission*

**SUMMARY:**

This study developed high-resolution land surface temperature data and agricultural drought indexes at the 8-day intervals, interpolating multiple remotely sensed data from the one-kilometer to 250-meter scales via co-kriging. The proposed drought indexes include Enhanced Vegetation Index, Vegetation Health Index (VHI), Temperature Condition Index (TCI), Soil Moisture Agricultural Drought Index (SMADI), and Normalized Vegetation Supply Water Index. Results show that NVSWI and SMADI captures well the spatial extent of droughts. Particularly, SMADI is a reliable drought index for the regions with water stress-sensitive croplands. This study concluded that these new developed hyper-resolution drought indexes provide a critical information for state-wise drought mitigation and land development policies.

The topic of this study is within the scope of *ESSD*. However, the data and methods are described in details. All the data and drought indexes developed in this study were not clearly summarized, which is confusing about the goal and expected outcomes from this study. Is the goal of this study to develop drought indices or vegetation indices? There are couple of major concerns about the proposed drought indexes. First, the temporal coverage of the proposed drought indices (2000–2017). Typically, 30 years are the minimum period for drought index calculation to capture realistic characteristics of droughts. Second, 10-centimeter soil layer moisture is a enough depth to capture propagation of soil moisture droughts. I am not sure that the 10-centimeter soil moisture layer has a moisture storage that can make a one-week delay response to drought in Figure 6. Was the delayed vegetation response found only at the one-week interval or longer intervals? In the manuscript, the delayed vegetation response to the on-going drought was highlighted but there is no explanation of potential physical processes that can make such a delayed vegetation response, for example, discussion of the possible role of vegetation types or land cover types.

Besides, key methods and equations for some indices (e.g., VHI and TCI) are missing. Without detailed descriptions of the methods, evaluations of the reproducibility of these data and indices are impossible. In addition, the manuscript was not written well with common grammatical errors and typos. Many parts of the manuscript are redundant. I think that the current manuscript is not publishable in *ESSD*, but encourage authors to consider resubmission of the manuscript adding detailed description of the methods and more informative interpretation of the results, and improving the writing. Here are the comments that I hope help improve the manuscript.

Line 14 and 16: The tense of these sentences are not consistent. Please use a consistent tense. This is a common grammatical error in this manuscript (see line 107-108, Line 126 & 128 & Line 148 & 150).

Please check the following reference site for Verb Tense Consistency: https://owl.purdue.edu/owl/general\_writing/grammar/verb\_tenses/verb\_tense\_consistency.ht ml

Line 15: Please use the full spelling, "meter", instead of "m".

Line 15: Please specify the drought indices that developed in this study.

Line 16: Please describe the temporal coverage of the developed dataset in the abstract (2000-2017).

Line 18: Please clarify what are the datasets (also line 22 and 23)? Are they drought indices? I felt that vegetation and drought indices were exchangeably used in this manuscript but they are not exchangeable. Drought indices are supposed to monitor water imbalance but vegetation indices are to monitor vegetation/crop growth. Therefore, vegetation indices can detect an adverse effect of severe droughts.

Line 40: Please delete "a". Another common grammatical errors.

Line 49: What are the difference between area affected and extent?

Line 55: the Palmer \*D\*rought Severity Index (PDSI). Please keep all the first letter of each in capital. A common editorial error (see Line 215-216).

Line 76: Please delete "high-resolution" for redundancy since the spatial resolution was described (0.05 degree). See another example in line 107-108.

Line 84-97 & Lien 98-106: They provide redundant information, which can be merged into one paragraph. Even I don't see losses of critical information by keeping one of the paragraphs.

Line 107: Please add "In this study,". What is a "high-resolution drought index"? Please specify it. It was not clear since several drought indices were also developed.

Line 107-108: These sentences should be written in a consistent tense. One of the most difficulties to follow this manuscript was so many indices and datasets, which makes me difficult to follow which drought indices are actually developed.

Line 136: Please use "coarse", instead of "limited".

Line 145: What does "combined" mean? Please describe how to combine LST and EVI. **Missing of description of the method.**

Line 156-157: Please delete "provides promising results since" since it is not necessary.

Line 158-161: These two sentences are very confusing. How come do two variable with a low correlation show a joint variability pattern?

Line 171-173: The sentences are confusing. What does "processes" mean? What are covariates here? Do they include SRTM? If yes, then SRTM was introduced as another covariate. Please clarify it and rewrite these sentences for clarification.

Line 190: Please describe how to obtain "VCI".

Line 194-195: What were "both indices"? Do they include VCI? VCI was obtained before TCI calculation, right? Please clarify it. Also, please describe clearly how to compute TCI. **Missing of description of the method.**

Line 198: Please specify an equation for VHI, instead of simply describing an additive combination of VCI and TCI. **Missing of description of the method.**

Line 206: Again, please specify how to combine the three variables. Missing of description of the method.

Line 215-216: How to compute NVSWI? Missing of description of the method.

Line 223: How to compute SEDI? What evaporation data used? The description of the SEDI calculation should be described briefly even though the reference was cited. **Missing of description of the method.**

Line 224: How to compute DSI?

Line 226: To evaluate high-resolution data, is it find to compare it with coarse-resolution data? Without doing it, it is obvious that high-resolution data show more detailed spatial heterogeneity. What about comparing the high-resolution data at the regional/state levels? High-resolution data might have more beneficial to capture droughts over mountainous regions rather than flat plain regions.

Line 240-241: Please specify how many diverse climatic settings and how to classify them (based on what?). **Missing of description of the method.**

Line 263: Again, are 12 years long enough to capture realistic characteristics of droughts?

Line 305: "identified \*a\* drought \*event in March 2002\*"

Line 307-308: Why was irrigation-driven alleviation of agriculture stress not found in DSI and SEDI? Is there any other potential causes instead of the coarse resolution? Simply comparing the coarse and high-resolution data does not fully support the claim that the irrigation-induced alleviation of water stress due to the coarse resolution.

Line 318-319: Please use "250-meter multiple drought index". Please use it consistently through the manuscript. See another example in Line 347-348

Line 330: decision and policymaking for what?

Line 330: Please use "This study", instead of "The current study".

Line 345: "helps to characterize ..."

Line 354: What does "separate" mean here?